# A High-Resolution Dataset for Instance Detection with Multi-View Instance Capture

**Qianqian Shen**[1,*]  **Yunhan Zhao**[2,*]  **Nahyun Kwon**[3]  **Jeeeun Kim**[3]  **Yanan Li**[1,†]  **Shu Kong**[3,4,5,†]

[1]Zhejiang Lab      [2]UC-Irvine      [3]Texas A&M University
[4]Institute of Collaborative Innovation      [5]University of Macau

shenqq@zhejianglab.com,   yunhaz5@ics.uci.edu,   {nahyunkwon, jeeeun.kim, shu}@tamu.edu,
liyn@zhejianglab.com,   skong@um.edu.mo

*Dataset and open-source code*

## Abstract

Instance detection (InsDet) is a long-lasting problem in robotics and computer vision, aiming to detect object instances (predefined by some visual examples) in a cluttered scene. Despite its practical significance, its advancement is overshadowed by Object Detection, which aims to detect objects belonging to some predefined classes. One major reason is that current InsDet datasets are too small in scale by today's standards. For example, the popular InsDet dataset GMU (published in 2016) has only 23 instances, far less than COCO (80 classes), a well-known object detection dataset published in 2014. We are motivated to introduce a new InsDet dataset and protocol. First, we define a realistic setup for InsDet: training data consists of multi-view instance captures, along with diverse scene images allowing synthesizing training images by pasting instance images on them with free box annotations. Second, we release a real-world database, which contains multi-view capture of 100 object instances, and high-resolution (6k×8k) testing images. Third, we extensively study baseline methods for InsDet on our dataset, analyze their performance and suggest future work. Somewhat surprisingly, using the off-the-shelf class-agnostic segmentation model (Segment Anything Model, SAM) and the self-supervised feature representation DINOv2 performs the best, achieving >10 AP better than end-to-end trained InsDet models that repurpose object detectors (e.g., FasterRCNN and RetinaNet).

## 1 Introduction

Instance detection (InsDet) requires detecting specific object instances (defined by some visual examples) from a scene image [12]. It is practically important in robotics, e.g., elderly-assistant robots need to fetch specific items (*my*-cup vs. *your*-cup) from a cluttered kitchen [40], micro-fulfillment robots for the retail need to pick items from mixed boxes or shelves [4].

**Motivation**. InsDet receives much less attention than the related problem of Object Detection (ObjDet), which aims to detect all objects belonging to some predefined classes [28, 37, 29, 48]. Fig. 1 compares the two problems. *One major reason is that there are not large-enough InsDet datasets by today's standards.* For example, the popular InsDet dataset GMU (published in 2016) [15] has only 23 object instances while the popular ObjDet dataset COCO has 80 object classes (published in 2014) [28]. Moreover, *there are no unified protocols in the literature of InsDet.* The current InsDet literature mixes multiple datasets to simulate training images and testing scenarios [12]. Note that the

---

*Equal contributions; †Corresponding authors.

37th Conference on Neural Information Processing Systems (NeurIPS 2023) Track on Datasets and Benchmarks.

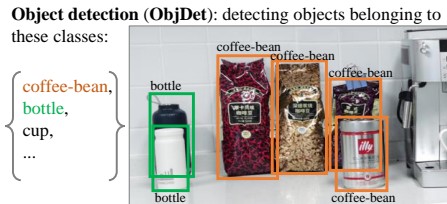 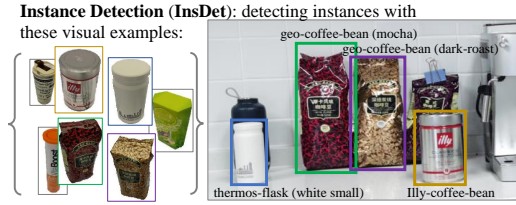

Figure 1: **Object detection (ObjDet) vs. instance detection (InsDet)**. ObjDet aims to detect all objects belonging to some predefined classes, whereas InsDet requires detecting specific object instances defined by some visual examples. Loosely speaking, InsDet treats a single object instance as a class compared to ObjDet. Please refer to Fig. 2-right for the challenge of InsDet, which is the focus of our work.

training protocol of InsDet does not follow that of ObjDet, which has training images annotated with bounding boxes. Differently, for InsDet[2], its setup should have profile images of instances (cf. right in Fig. 1) and optionally diverse background images not containing such instances [12]. We release a new dataset and present a unified protocol to foster the InsDet research.

**Overview of our dataset** is presented in Fig. 2. In our dataset, profile images (3072x3072) of object instances and testing images (6144x8192) are high-resolution captured by a Leica camera (commonly used in today's cellphones). This inexpensive camera is deployable in current or future robot devices. Hence, our dataset simulates real-world scenarios, e.g., robotic navigation in indoor scenes. Even with high-resolution images, objects in testing images appear small, taking only a tiny region in the high-res images. This demonstrates a clear challenge of InsDet in our dataset. Therefore, our dataset allows studying InsDet methods towards real-time operation on high-res (as future work).

**Preview of technical insights**. On our dataset, we revisit existing InsDet methods [26, 12, 16]. Perhaps the only InsDet framework is cut-paste-learn [12], which cuts instances from their profile images, pastes them on random background images (so being able to derive "free" bounding boxes annotations), and trains InsDet detectors on such data by following that of ObjDet (e.g., Faster-RCNN [37]). We study this framework, train different detectors, and confirm that the state-of-the-art transformer-based detector DINO [48] performs the best, achieving 27.99 AP, significantly better than CNN-based detector FasterRCNN (19.52 AP). Further, we present a non-learned method that runs off-the-shelf proposal detectors (SAM [23] in our work) to generate object proposals and use self-supervised learned features (DINO$_f$ [8][3] and DINOv2$_f$ [33]) to find matched proposals to instances' profile images. Surprisingly, this non-learned method resoundingly outperforms end-to-end learning methods, i.e., SAM+DINOv2$_f$ achieves 41.61 AP, much better than DINO (27.99 AP) [48].

**Contributions**. We make three major contributions.

1. We formulate the InsDet problem with a unified protocol and release a challenging dataset consisting of both high-resolution profile images and high-res testing images.
2. We conduct extensive experiments on our dataset and benchmark representative methods following the cut-paste-learn framework [12], showing that stronger detectors perform better.
3. We present a non-learned method that uses an off-the-shelf proposal detector (i.e., SAM [23]) to produce proposals, and self-supervised learned features (e.g., DINOv2$_f$ [33]) to find instances (which are well matched to their profile images). This simple method significantly outperforms the end-to-end InsDet models.

## 2 Related Work

**Instance detection (InsDet)** is a long-lasting problem in computer vision and robotics [49, 12, 32, 3, 15, 21, 4], referring to detecting specific object instances in a scene image. Traditional InsDet methods use keypoint matching [34] or template matching [19]; more recent ones train deep neural networks to approach InsDet [32]. Some others focus on obtaining more training samples by rendering realistic instance examples [22, 21], data augmentation [12], and synthesizing training images by cutting

---

[2]In real-world applications (e.g., robot learning), it is infeasible to place objects in diverse scenes, take scene photos, then annotate instances using boxes towards training images (cf. training data in object detection).

[3]We add subscript $_f$ to indicate that DINO$_f$ [8] is the self-supervised learned feature extractor; distinguishing it from a well-known object detector DINO [48].

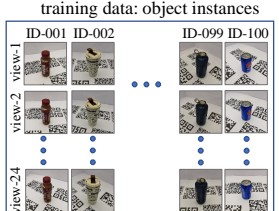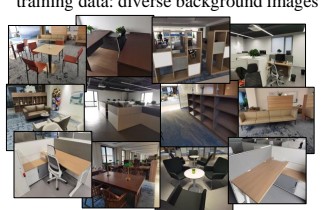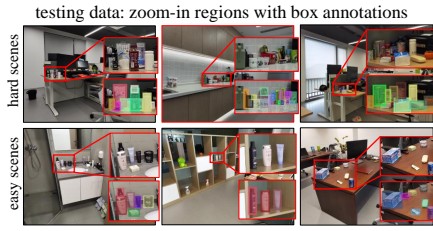

training data: object instances     training data: diverse background images     testing data: zoom-in regions with box annotations

Figure 2: **Overview of our instance detection dataset**. **Left**: It contains 100 distinct object instances. For each of them, we capture 24 profile photos from multiple views. We paste QR code images beneath objects to allow relative camera estimation (e.g., by COLMAP [41]), just like other existing datasets [20, 5]. **Middle**: We take photos in random scenes (which do not contain any of the 100 instances) as background images. The background images can be optionally used to synthesize training data, e.g., pasting the foreground instances on them towards box-annotated training images [26, 12, 16] as used in the object detection literature [28]. **Right**: high-resolution (6k×8k) testing images of clutter scenes contain diverse instances, including some of the 100 predefined instances and other uninterested ones. The goal of InsDet is to detect the predefined instances in these testing images. From the zoom-in regions, we see the scene clutters make InsDet a rather challenging problem.

instances as foregrounds and pasting them to background images [26, 12, 16]. Speaking of InsDet datasets, [15] collects scene images from 9 kitchen scenes with RGB-D cameras and defines 23 instances of interest to annotate with 2D boxes on scene images; [21] creates 3D models of 29 instances from 6 indoor scenes, and uses them to synthesize training and testing data; [4] creates 3D mesh models of 100 grocery store objects, renders 80 views of images for each instance, and uses them to synthesize training data.

As for benchmarking protocol of InsDet, [12] synthesizes training data from BigBird [43] and UW Scenes [25] and tests on the GMU dataset [15]; [21] trains on their in-house data and test on LM-O [5] and Rutgers APC [38] datasets. Moreover, some works require hardware-demanding setups [4], some synthesize both training and testing data [21, 26], while others mix existing datasets for benchmarking [12]. Given that the modern literature on InsDet lacks a unified benchmarking protocol (till now!), we introduce a more realistic unified protocol along with our InsDet dataset, allowing fairly benchmarking methods and fostering research of InsDet.

**Object detection (ObjDet)** is a fundamental computer vision problem [13, 28, 37], requiring detecting all objects belonging to some predefined categories. The prevalent ObjDet detectors adopt convolutional neural networks (CNNs) as a backbone and a detector-head for proposal detection and classification, typically using bounding box regression and a softmax-classifier. Approaches can be grouped into two categories: one-stage detectors [36, 30, 35, 46] and two-stage detectors [17, 6]. One-stage detectors predict candidate detection proposals using bounding boxes and labels at regular spatial positions over feature maps; two-stage detectors first produce detection proposals, then perform classification and bounding box regression for each proposal. Recently, the transformer-based detectors transcend CNN-based detectors [7, 51, 48], yielding much better performance on various ObjDet benchmarks. Different from ObjDet, InsDet requires distinguishing individual object instances within a class. Nevertheless, to approach InsDet, the common practice is to repurpose ObjDet detectors by treating unique instances as individual classes. We follow this practice and benchmark various ObjDet methods on our InsDet dataset.

**Pretrained models**. Pretraining is an effective way to learn features from diverse data. For example, training on the large-scale ImageNet dataset for image classification [10], a neural network can serve as a powerful feature extractor for various vision tasks [11, 42]. Object detectors trained on the COCO dataset [28] can serve as a backbone allowing finetuning on a target domain to improve detection performance [27]. Such pretraining requires human annotations which can be costly. Therefore, self-supervised pretraining has attracted increasing attention and achieved remarkable progress [9, 18, 8, 33]. Moreover, the recent literature shows that pretraining on much larger-scale data can serve as a foundation model for being able to perform well across domains and tasks. For example, the Segment Anything Model (SAM) pretrains a class-agnostic proposal detector on web-scale data and shows an impressive ability to detect and segment diverse objects in the wild [23]. In this work, with our high-res InsDet dataset, we explore a non-learned method by using publicly available pretrained models. We show that such a simple method significantly outperforms end-to-end learned InsDet detectors.

# 3   Instance Detection: Protocol and Dataset

In this section, we formulate a realistic unified InsDet protocol and introduce the new dataset. We release our dataset under the MIT License, hoping to contribute to the broader research community.

## 3.1   The Protocol

Our InsDet protocol is motivated by real-world indoor robotic applications. In particular, we consider the scenario that assistive robots must locate and recognize instances to fetch them in a cluttered indoor scene [40], where InsDet is a crucial component. Realistically, for a given object instance, the robots should see it only from a few views (*at the training stage*), and then accurately detect it *in a distance* in *any* scenes (*at the testing stage*). Therefore, we suggest the protocol specifying the training and testing setups below. We refer the readers to Fig. 2 for an illustration of this protocol.

- **Training**. There are profile images of each instance captured at different views and diverse background images. The background images can be used to synthesize training images with free 2D-box annotations, as done by the cut-paste-learn methods [26, 12, 16].
- **Testing**. InsDet algorithms are required to precisely detect all predefined instances from real-world images of cluttered scenes.

**Evaluation metrics**. The InsDet literature commonly uses average precision (AP) at IoU=0.5 [12, 2, 32]; others use different metrics, e.g., AP at IoU=0.75 [21], mean AP [3, 15], and F1 score [4]. As a single metric appears to be insufficient to benchmark methods, we follow the literature of ObjDet that uses multiple metrics altogether [28].

- **AP** averages the precision at IoU thresholds from 0.5 to 0.95 with the step size 0.05. It is the *primary metric* in the most well-known COCO Object Detection dataset [28].
- $AP_{50}$ and $AP_{75}$ are the precision averaged over all instances with IoU threshold as 0.5 and 0.75, respectively. In particular, $AP_{50}$ is the widely used metric in the literature of InsDet.
- **AR** (average recall) averages the proposal recall at IoU threshold from 0.5 to 1.0 with the step size 0.05, regardless of the classification accuracy. AR measures the localization performance (excluding classification accuracy) of an InsDet model.

Moreover, we tag *hard* and *easy* scenes in the testing images based on the level of clutter and occlusion, as shown by the right panel of Fig. 2.

## 3.2   The Dataset

We introduce a challenging real-world dataset of indoor scenes (motivated by indoor assistive robots), including high-resolution photos of 100 distinct object instances, and high-resolution testing images captured from 14 indoor scenes where there are such 100 instances defined for InsDet. Table 1 summarizes the statistics compared with existing datasets, showing that our dataset is larger in scale and more challenging than existing InsDet datasets. Importantly, object instances are located far from the camera in cluttered scenes; this is realistic because robots must detect objects in a distance before approaching them [1]. Perhaps surprisingly, only a few InsDet datasets exist in the literature. Among them, Grocery [4], which is the latest and has the most instances like our dataset, is not publicly available.

Our InsDet dataset contains 100 object instances. When capturing photos for each instance, inspired by prior arts [43, 20, 5], we paste a QR code on the tabletop, which enables pose estimation, e.g., using COLMAP [41]. Yet, we note more realistic scenarios can be hand-holding instances for capturing [24], which we think of as future work. In Fig. 3, we plot the per-instance frequency in the testing set. Each instance photo is of 3072×3072 pixel resolution. For each instance, we capture 24 photos from multiple views. The left panel of Fig. 2 shows some random photos for some instances. For the testing set, we capture high-resolution images (6144×8192) in cluttered scenes, where some instances are placed in reasonable locations, as shown in the right panel of Fig. 2. We tag these images as *easy* or *hard* based on scene clutter and object occlusion levels. When objects are placed sparsely, we tag the testing images as *easy*; otherwise, we tag them as *hard*. Our InsDet dataset also contains 200 high-res background images of indoor scenes (cf. Fig. 2-middle). These indoor scenes are not included in testing images. They allow using the cut-paste-learn framework to synthesize training images [26, 12, 16]. Following this framework, we segment foreground instances using GrabCut [39]

Table 1: **Comparison of our dataset to existing ones**. Several datasets are used in the InsDet literature although they are designed for different tasks. For example, BigBird and LM are designed to study algorithms of object recognition and object pose estimation, hence they contain instances that are close to the camera. Naively repurposing them for InsDet leads to saturated performance, impoverishing the exploration space of InsDet. Instead, ours is more challenging as instances are placed far from the camera, simulating realistic scenarios where robots must detect instances at a distance. Importantly, our dataset contains far more instances than other publicly available InsDet datasets.

| | for what task | publicly available | #instances | #scenes | published year | resolution |
|---|---|---|---|---|---|---|
| BigBird [43] | recognition | ✓ | 100 | N/A | 2014 | 1280x1024 |
| RGBD [26] | scene label. | ✓ | 300 | 14 | 2017 | N/A |
| LM [20] | 6D pose est. | ✓ | 15 | 1 | 2012 | 480x640 |
| LM-O [5] | 6D pose est. | ✓ | 20 | 1 | 2017 | 480x640 |
| RU-APC [38] | 3D pose est. | ✓ | 14 | 1 | 2016 | 480x640 |
| GMU [15] | InsDet | ✓ | 23 | 9 | 2016 | 1080x1920 |
| AVD [1] | InsDet | ✓ | 33 | 9 | 2017 | 1080x1920 |
| Grocery [4] | InsDet | ✗ | 100 | 10 | 2021 | unknown |
| Ours | InsDet | ✓ | 100 | 14 | 2023 | 6144x8192 |

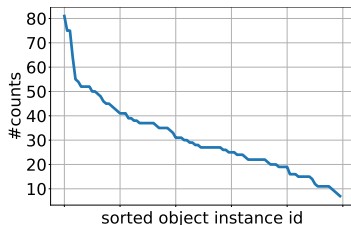

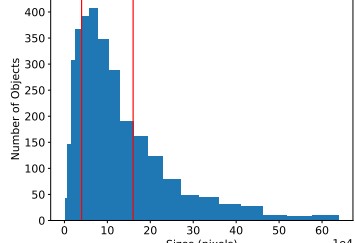

| size | bounding box area |
|---|---|
| small | $< 200^2$ |
| medium | $200^2$ - $400^2$ |
| large | $> 400^2$ |

Figure 3: Imbalanced distribution of instances in test-set. Yet, instances have the same number of profile images in training and the metrics average over all instances. So, the evaluation is unbiased.

Table 2: Following the spirit of COCO dataset, we tag objects with different sizes by `small`, `medium`, and `large`, respectively.

Figure 4: Distribution of objects w.r.t their bounding box area in testing images. We split them into `small`, `medium`, and `large` subgroups to allow breakdown analysis.

to paste them on background images. It is worth noting that the recent vision foundation model SAM [23] makes interactive segmentation much more efficient. Yet, this work is made public after we collected our dataset. Following the COCO dataset [28], we further tag testing object instances as *small*, *medium*, and *large* according to their bounding box area, as in Table 2. To determine their size tags, we plot the distribution of their sizes in Fig. 4, showing an intuitive way to tag them.

## 4 Methodology

### 4.1 The Strong Baseline: Cut-Paste-Learn

**Cut-Paste-Learn** serves as a strong baseline that synthesizes training images with 2D-box annotations [12]. This allows one to train InsDet detectors in the same way as training normal ObjDet detectors, by simply treating the $K$ unique instances as $K$ distinct classes. It cuts and pastes foreground instances at various aspect ratios and scales on diverse background images, yielding synthetic training images, as shown in Fig. 5. Cut-paste-learn is model-agnostic, allowing one to adopt any state-of-the-art detector architecture. In this work, we study five popular detectors, covering the two-stage detector FasterRCNN [37], and one-stage anchor-based detector RetinaNet [29], and one-stage anchor-free detectors CenterNet [49], and FCOS [45]; and the transformer-based detector DINO [48]. There are multiple factors in the cut-paste-learn framework, such as the number of inserted objects in each background image, their relative size, the number of generated training images and blending methods. We conduct comprehensive ablation studies and report results using the best-tuned choices. We refer interested readers to the supplement for the ablation studies.

| (a) Box | (b) Gaussian blurring | (c) Motion | (d) Naive pasting |

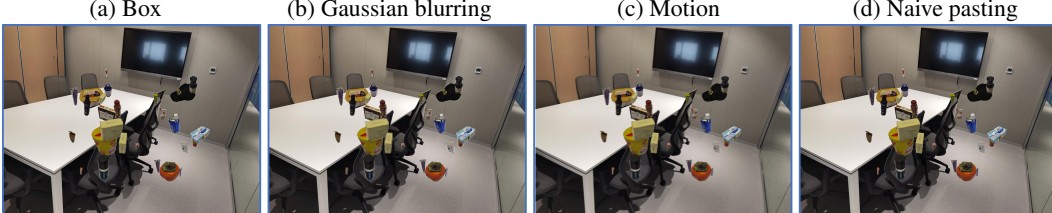

Figure 5: Synthetic training images for cut-paste-learn methods. We use different blending methods to paste object instances on the same background. We recommend that interested readers refer to the supplement for an ablation study using different blending methods.

## 4.2   The Simple, Non-Learned Method

We introduce a simple, non-learned InsDet method by exploiting publicly available pretrained models. This method consists of three main steps: (1) proposal generation on testing images, (2) matching proposals and profile images, (3) selecting the best-matched proposals as the detected instances.

**Proposal generation**. We use the recently released Segment Anything Model (SAM) [23] to generate proposals. For a proposal, we define a minimum bounding square box encapsulating the masked instance, and then crop the region from the high-resolution testing image. SAM not only achieves high recall (Table 4) on our InsDet dataset but detects objects not belonging to the instances of interest. So the next step is to find interested instances from the proposals.

**Feature representation of proposals and profile images**. Intuitively, among the pool of proposals, we are interested in those that are well-matched to any profile images of any instance. The well-matched ones are more likely to be predefined instances. To match proposals and profile images, we use off-the-shelf features to represent them. In this work, we study two self-supervised learned models as feature extractors, i.e. $DINO_f$ [8], and $DINOv2_f$ [33]. We feed a square crop (of a proposal) or a profile image to the feature extractor to obtain its feature representation. We use cosine similarity over the features as the similarity measure between a proposal and a profile image.

**Proposal matching and selection**. As each instance has multiple profile images, we need to design the similarity between a proposal and an instance. For a proposal, we compute the cosine similarities of its feature to all the profile images of an instance and use the maximum as its final similarity to this instance. We then filter out proposals and instances if they have similarities lower than a threshold, indicating that they are not matched to any instances or proposals. Finally, we obtain a similarity matrix between all remaining proposals and all remaining instances. Over this matrix, we study two matching algorithms to find the best match (hence the final InsDet results), i.e. Rank & Select, and Stable Matching [14, 31]. The former is a greedy algorithm that iteratively selects the best match (highest cosine similarity) between a proposal and an instance and removes the corresponding proposal until no proposal/instance is left. The latter produces an optimal list of matched proposals and instances, such that there exist no pair of instances and proposals which both prefer each other to their current correspondence under the matching.

## 5   Experiments

**Synthesizing training images for cut-paste-learn baselines**. Our baseline method trains state-of-the-art ObjDet detectors on data synthesized using the cut-paste-learn strategy [12]. For evaluating on our InsDet dataset, we generate 19k training examples and 6k validation examples. For each example, various numbers of foreground objects ranging from 25 to 35 are pasted to a randomly selected background image. The objects are randomly resized with a scale from 0.15 to 0.5. We use four blending options [12], including Gaussian blurring, motion blurring, box blurring, and naive pasting. Fig. 5 shows some random synthetic images. The above factors have a notable impact on the final performance of trained models, and we have conducted a comprehensive ablation study. We refer interested readers to the supplement for the study.

**Implementation details.** We conduct all the experiments based on open-source implementations, such as Detectron2 [47] (for FasterRCNN and RetinaNet), CenterNet [50], FCOS [44] and DINO [48]. The CNN-based end-to-end detectors are initialized with pretrained weights on COCO [28]. We fine-tune CNN-based models using SGD and the transformer-based model using AdamW with a

Table 3: **Benchmarking results on our dataset**. We summarize three salient conclusions. (1) End-to-end trained detectors perform better with stronger detector architectures, e.g., the transformer DINO (27.99 AP) outperforms FasterRCNN (19.54 AP). (2) Interestingly, the non-learned method SAM+DINOv2$_f$ performs the best (41.61 AP), significantly better than end-to-end learned detectors including DINO (27.99 AP). (3) All methods have much lower AP on `hard` testing images or `small` objects (e.g., SAM+DINOv2$_f$ yields 28.03 AP on `hard` vs. 47.57 AP on `easy`), showing that future work should focus on `hard` situations or `small` instances.

| | **AP** | | | | | | **AP$_{50}$** | **AP$_{75}$** |
|---|---|---|---|---|---|---|---|---|
| | avg | hard | easy | small | medium | large | | |
| FasterRCNN [37] | 19.54 | 10.26 | 23.75 | 5.03 | 22.20 | 37.97 | 29.21 | 23.26 |
| RetinaNet [29] | 22.22 | 14.92 | 26.49 | 5.48 | 25.80 | 42.71 | 31.19 | 24.98 |
| CenterNet [49] | 21.12 | 11.85 | 25.70 | 5.90 | 24.15 | 40.38 | 32.72 | 23.60 |
| FCOS [45] | 22.40 | 13.22 | 28.68 | 6.17 | 26.46 | 38.13 | 32.80 | 25.47 |
| DINO [48] | 27.99 | 17.89 | 32.65 | 11.51 | 31.60 | 48.35 | 39.62 | 32.19 |
| SAM + DINO$_f$ | 36.97 | 22.38 | 43.88 | 11.93 | 40.85 | 62.67 | 44.13 | 40.42 |
| SAM + DINOv2$_f$ | **41.61** | **28.03** | **47.57** | **14.58** | **45.83** | **69.14** | **49.10** | **45.95** |

Table 4: **Benchmarking results w.r.t average recall (AR) for *small*, *medium* and *large* instances**. "AR@max10" means AR within the top-10 ranked detections. In computing AR, we rank detections by using the detection confidence scores of the learning-based methods (e.g., FasterRCNN) or similarity scores in the non-learned methods (e.g., SAM+DINO$_f$). AR$_s$, AR$_m$, and AR$_l$ are breakdowns of AR for small, medium and large testing object instances. Results show that (1) the non-learned methods that use SAM generally recall more instances than others, and (2) all methods suffer from small instances. In sum, results show that methods yielding higher recall achieve higher AP metrics (cf. Table 3).

| | **AR@max10** | **AR@max100** | **AR$_s$@max100** | **AR$_m$@max100** | **AR$_l$@max100** |
|---|---|---|---|---|---|
| FasterRCNN [37] | 26.24 | 39.24 | 14.83 | 44.87 | 60.05 |
| RetinaNet [29] | 26.33 | 49.38 | 22.04 | 56.76 | 69.69 |
| CenterNet [49] | 23.55 | 44.72 | 17.84 | 52.03 | 64.58 |
| FCOS [45] | 25.82 | 46.28 | 22.09 | 52.85 | 64.11 |
| DINO [48] | 29.84 | 54.22 | **32.00** | 59.43 | 72.92 |
| SAM + DINO$_f$ | 31.25 | 63.05 | 31.65 | 70.01 | **90.63** |
| SAM + DINOv2$_f$ | **40.02** | **63.06** | 31.11 | **70.40** | 90.36 |

learning rate of 1e-3 and a batch size of 16. We fine-tune all the models for 5 epochs (which are enough for training to converge) and evaluate checkpoints after each epoch for model selection. The models are trained on a single Tesla V100 GPU with 32G memory.

If applied, we preprocess object instance profile images and proposals. Specifically, for a profile image, we remove the background pixels (e.g., pixels of QR code) using foreground segmentation (i.e., GrabCut). For each proposal, we crop its minimum bounding square box. We also study whether removing background pixels by using SAM's mask output performs better. We use DINO$_f$ and DINOv2$_f$ to compute feature representations.

## 5.1 Benchmarking Results

**Quantitative results**. To evaluate the proposed InsDet protocol and dataset, we first train detectors from a COCO-pretrained backbone following the cut-past-learn baseline. Table 3 lists detailed comparisons and Fig. 6 plots the precision-recall curves for the compared methods. We can see that detectors with stronger architectures perform better, e.g. DINO (27.99% AP) vs. FasterRCNN (19.54% AP). Second, non-learned methods outperform end-to-end trained models, e.g., SAM+DINOv2$_f$ (41.61% AP) vs. DINO (27.99% AP). Third, all the methods perform poorly on *hard* scenes and *small* instances, suggesting future work focusing on such cases.

Table 4 compares methods w.r.t the average recall (AR) metric. "AR@max10" means AR within the top-10 ranked detections. In computing AR, we rank detections by using the detection confidence scores of the learning-based methods (e.g., FasterRCNN) or similarity scores in the non-learned methods (e.g., SAM+DINO$_f$). AR$_s$, AR$_m$, and AR$_l$ are breakdowns of AR for small, medium, and large testing object instances. Results show that (1) the non-learned methods that use SAM generally recall more instances than others, and (2) all methods suffer from small instances. In sum, results show that methods yielding higher recall achieve higher AP metrics (cf. Table 3). Table 5 further studies AR in *hard* and *easy* scenes. We can observe that: (1) the non-learned methods that use SAM

Table 5: **Benchmarking results w.r.t average recall (AR) for *hard* and *easy* scenes**. We add a breakdown analysis of testing images on `hard` and `easy` scenes. Results show that (1) the non-learned methods that use SAM generally recall more instances than others, and (2) all methods suffer from hard scenes.

| | AR@max10 | | | AR@max100 | | |
|---|---|---|---|---|---|---|
| | avg | hard | easy | avg | hard | easy |
| FasterRCNN [37] | 26.24 | 12.92 | 32.33 | 39.24 | 16.91 | 49.43 |
| RetinaNet [29] | 26.33 | 15.38 | 31.33 | 49.38 | 29.00 | 58.69 |
| CenterNet [49] | 23.55 | 11.87 | 28.87 | 44.72 | 24.88 | 53.76 |
| FCOS [45] | 25.82 | 12.81 | 31.74 | 46.28 | 26.55 | 55.27 |
| DINO [48] | 29.84 | 16.63 | 35.84 | 54.22 | 36.46 | 62.30 |
| SAM + DINO$_f$ | 31.25 | 16.96 | 37.73 | 63.05 | 42.46 | **72.41** |
| SAM + DINOv2$_f$ | **40.02** | **27.64** | **45.36** | **63.06** | **43.47** | 71.96 |

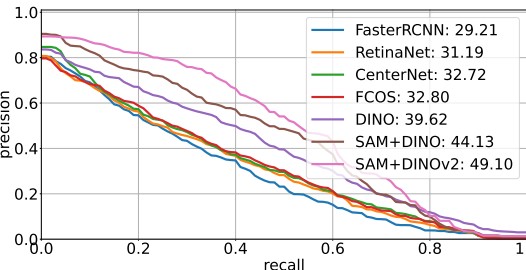

Figure 6: Precision-recall curves with IoU=0.5 (AP50 in the legend) on our InsDet dataset. Stronger detectors perform better, e.g., DINO, a transformer-based detector significantly outperforms FasterRCNN. Furthermore, even with a simple non-learned method, leveraging pretrained models, e.g., SAM+DINOv2$_f$, outperforms end-to-end learned methods.

recall more proposals than other competitors in both *hard* and *easy* scenes; (2) all methods basically suffer from *hard* scenes.

**Qualitative results**. Fig. 7 visualizes qualitative results on two testing examples from the InsDet dataset. Stronger detectors, e.g., the non-learned method SAM+DINOv2$_f$, produce fewer false negatives. Even so, all detectors still struggle to detect instances with presented barriers such as heavy occlusion, instance size being too small, etc. As shown in Fig. 6, the non-learned method SAM+DINOv2$_f$ outperforms end-to-end learned methods in a wide range of recall thresholds.

## 5.2 Ablation Study

Due to the space limit, we ablate the instance crop and stable matching in the main paper and put more (including ablation studies for the cut-paste-learn methods) in the supplement.

**Proposal feature extraction in the non-learned method.** Given a box crop (encapsulating the proposal) generated by SAM in the non-learned method, we study how to process the crop to improve InsDet performance. Here, we can either crop and feed its minimum bounding box to compute DINOv2$_f$ features, or we can use the mask to remove the background in the box. Table 6 shows the comparison. Clearly, the latter performs remarkably better in both *hard* and *easy* scenarios.

**Proposal-instance match in the non-learned method.** After generating proposals by SAM, we need to compare them with instance profile images to get the final detection results. We study the InsDet performance of the two matching algorithms. Rank & Select is a greedy algorithm that iteratively finds the best match between any proposals and instances until no instances/proposals are left unmatched; stable matching produces an optimal list of matched proposals and instances such that there does not exist a pair in which both prefer other proposals/instances to their current correspondence under the matching. Table 7 compares these two methods, clearly showing that stable matching works better.

**Impact of different image/proposal resolutions.** We study the InsDet performance when using images of different sizes for SAM, and using different resolutions of crops fed into DINOv2$_f$. For example, we can use SAM on images of size 3072×4096 to generate proposals, we can resize crops of proposals to 224×224 to feed into DINOv2$_f$ if they are larger than 224×224 (otherwise, keep them unchanged). Table 8 lists detailed comparisons. We have two observations. (1) The InsDet performance generally increases with the image resolution but starts to drop when the input image is too large, i.e., 6144×8192. This is because SAM tends to produce object parts as individual instances,

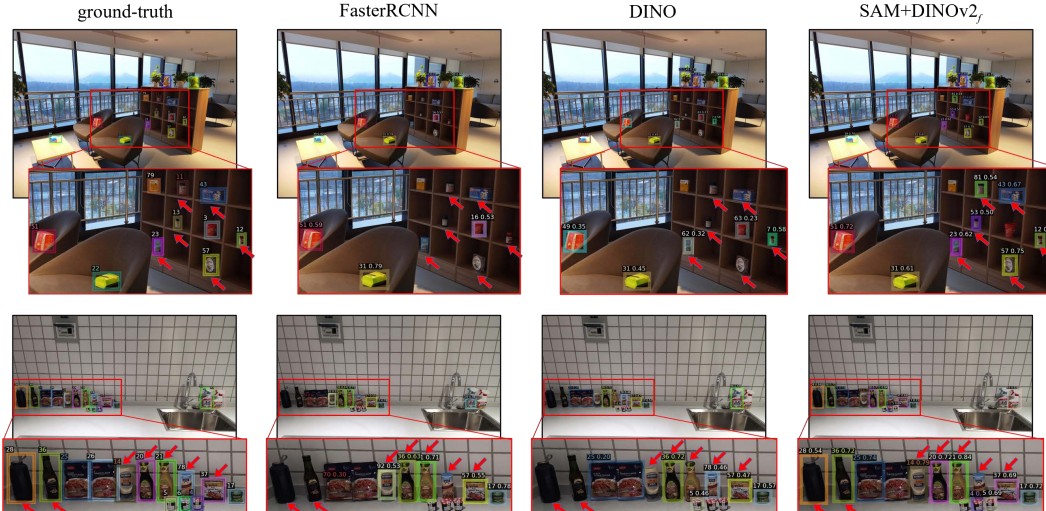

Figure 7: Visual results of FasterRCNN, DINO, and SAM+DINOv2$_f$ on our InsDet dataset. The top row illustrates the sparse placement of instances (i.e., `easy` scenario), while the bottom contains more cluttered instances (i.e., `hard` scenario). We drop predicted instance names for brevity. SAM helps localize instances with more precise bounding boxes, e.g., as arrows labeled in the upper row. DINOv2$_f$ provides more precise recognition of localized instances, e.g., five instances in the right of the bottom row. Compared with DINO, SAM+DINOv2$_f$ is better at locating occluded instances.

Table 6: **Ablation study: whether to remove background in crops for feature computation**. Based on a proposal given by SAM, we can crop and feed its minimum bounding square to compute DINOv2$_f$ feature, or we can use the mask to remove the background in the square before computing the feature. Clearly, the latter performs remarkably better.

| strategy | **AP** | | | **AP$_{50}$** | | | **AP$_{75}$** | | |
|---|---|---|---|---|---|---|---|---|---|
| | avg | hard | easy | avg | hard | easy | avg | hard | easy |
| w/o background removal | 36.04 | 23.04 | 42.37 | 43.84 | 29.12 | 51.00 | 39.59 | 25.74 | 46.13 |
| w/ background removal | 39.12 | 24.00 | 47.17 | 46.72 | 30.81 | 54.66 | 42.86 | 26.40 | 51.58 |

Table 7: **Ablation study: whether to generate unique proposal-instance match**. In contrast to Rank&Select, Stable Matching produces a unique match to proposal/instance for each instance/proposal, yielding better performance than Rank&Select.

| strategy | **AP** | | | **AP$_{50}$** | | | **AP$_{75}$** | | |
|---|---|---|---|---|---|---|---|---|---|
| | avg | hard | easy | avg | hard | easy | avg | hard | easy |
| Rank & Select | 38.62 | 23.95 | 46.31 | 46.04 | 30.77 | 53.64 | 42.37 | 26.39 | 50.61 |
| Stable Matching | 39.12 | 24.00 | 47.17 | 46.72 | 30.81 | 54.66 | 42.86 | 26.40 | 51.58 |

Table 8: **Ablation study: which image/proposal size to use for SAM and DINOv2$_f$.** We notice that InsDet performance generally increases with the input image resolution, but starts to drop when the image is too large. When using larger proposals for DINOv2$_f$, InsDet performance also gets better.

| image resolution for SAM | input size for DINOv2$_f$ | **AP** | | | **AP$_{50}$** | | | **AP$_{75}$** | | |
|---|---|---|---|---|---|---|---|---|---|---|
| | | avg | hard | easy | avg | hard | easy | avg | hard | easy |
| 768×1024 | | 36.46 | 21.67 | 43.88 | 46.22 | 29.52 | 54.11 | 41.53 | 24.95 | 49.99 |
| 1536×2048 | 224×224 | 39.12 | 24.00 | 47.17 | 46.72 | 30.81 | 54.66 | 42.86 | 26.40 | 51.58 |
| 3072×4096 | | 39.17 | 24.08 | 46.60 | 45.71 | 30.34 | 53.07 | 41.90 | 26.12 | 49.71 |
| 6144×8192 | | 38.74 | 23.39 | 46.29 | 45.24 | 29.24 | 52.78 | 40.81 | 25.14 | 48.65 |
| | 112×112 | 26.46 | 16.52 | 31.32 | 30.83 | 20.94 | 36.26 | 28.89 | 18.81 | 33.73 |
| 1536×2048 | 224×224 | 39.12 | 24.00 | 47.17 | 46.72 | 30.81 | 54.66 | 42.86 | 26.40 | 51.58 |
| | 448×448 | 41.61 | 28.03 | 47.57 | 49.10 | 36.64 | 54.84 | 45.95 | 31.41 | 52.03 |

| method | time (sec) | AP (%) |
|---|---|---|
| FasterRCNN [37] | 0.00399 | 19.54 |
| RetinaNet [29] | 0.00412 | 22.22 |
| CenterNet [49] | 0.00376 | 21.12 |
| FCOS [45] | 0.00271 | 22.40 |
| DINO [48] | 1.90625 | 27.99 |
| SAM + DINO$_f$ | 15.10 | 36.97 |
| SAM + DINOv2$_f$ | 14.70 | 41.61 |

Table 9: We compare the inference runtime (second/image) of different methods, along with their InsDet performance in AP (%). Clearly, there is a trade-off between runtime and detection precision. For example, among the methods studied in our work, SAM+DINOv2$_f$ achieves the highest AP (41.61) but is four orders of magnitude slower than FasterRCNN (19.54 AP). Developing faster and better InsDet methods is apparently future work.

resulting in more false positives. (2) When using larger proposals for DINOv2$_f$, InsDet performance gets better, e.g., 41.61% (448×448) vs. 39.12% (224×224).

## 5.3 Runtime Comparison

Table 9 compares the runtime of different methods, along with their InsDet performance. There is a trade-off between runtime and detection precision. For example, among the methods studied in our work, SAM+DINOv2$_f$ achieves the highest AP (41.61) but is four orders of magnitude slower than FasterRCNN (19.54 AP). Developing faster and better InsDet methods is future work.

## 5.4 Discussions

**Societal impact**. InsDet is a crucial component in various robotic applications such as elderly-assistive agents. Hence, releasing a unified benchmarking protocol contributes to broader communities. While our dataset enables InsDet research to move forward, similar to other works, directly applying algorithms brought by our dataset is risky in real-world applications.

**Limitations**. We note several limitations in our current work. First, while our work uses normal cameras to collect datasets, we expect to use better and cheaper hardware (e.g., depth camera and IMU) for data collection. Second, while the cut-paste-learn method we adopt does not consider geometric cues when synthesizing training images, we hope to incorporate such information to generate better and more realistic training images, e.g., pasting instances only on up-surfaces like tables, desks, and floors. Third, while SAM+DINOv2$_f$ performs the best, this method is time-consuming (see a run-time study in the supplement); real-world applications should consider real-time requirements.

**Future work**. In view of the above limitations, the future work includes: (1) Exploring high-resolution images for more precise detection on *hard* situations, e.g., one can combine proposals generated from multi-scale and multi-resolution images. (2) Developing faster algorithms, e.g., one can use multi-scale detectors to attend to regions of interest for progressive detection. (3) Bridging end-to-end fast models and powerful yet slow pretrained models, e.g., one can train lightweight adaptors atop pretrained models for better InsDet.

## 6 Conclusion

We explore the problem of Instance Detection (InsDet) by introducing a new dataset consisting of high-resolution images and formulating a realistic unified protocol. We revisit representative InsDet methods in the cut-paste-learn framework and design a non-learned method by leveraging publicly-available pretrained models. Extensive experiments show that the non-learned method significantly outperforms end-to-end InsDet models. Yet, the non-learned method is slow because running large pretrained models takes more time than end-to-end trained models. Moreover, all methods struggle in hard situations (e.g., in front of heavy occlusions and a high level of clutter in the scene). This shows that our dataset serves as a challenging venue for the community to study InsDet.

## Acknowledgements

This work is supported by NSFC (No.62206256), and University of Macau (SRG2023-00044-FST). Shu Kong acknowledges Dr. Bin Liu for the initial support via compute resource.

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
