# A High-Resolution Dataset for Instance Detection with Multi-View Instance Capture
## *(Supplemental Material)*

**Qianqian Shen**[1,*] **Yunhan Zhao**[2,*] **Nahyun Kwon**[3] **Jeeeun Kim**[3] **Yanan Li**[1,†] **Shu Kong**[3,4,5,†]

[1]Zhejiang Lab       [2]UC-Irvine       [3]Texas A&M University
[4]Institute of Collaborative Innovation       [5]University of Macau
shenqq@zhejianglab.com,   yunhaz5@ics.uci.edu,   {nahyunkwon, jeeeun.kim, shu}@tamu.edu,
liyn@zhejianglab.com,   skong@um.edu.mo

*Dataset and open-source code*

## Outline

This document supplements the main paper with more experimental results, more visualizations, further details of the InsDet dataset, and open-source code. Below is the outline of this document.

- **Section 1**. We provide demo code for the non-learned method using Jupyter Notebook.
- **Section 2**. We visualize object instance profile images captured in multiple views, instance proposals produced by SAM, and input crops to $DINO_f$ / $DINOv2_f$.
- **Section 3**. We conduct extensive ablation studies on the traditional cut-paste-learn method.
- **Section 4** includes dataset documentation and intended uses.
- **Appendix** contains further details of dataset collection.

## 1   Open-Source Code

We release open-source code in the form of Jupyter Notebook plus Python files.

**Why Jupyter Notebook?** We prefer to release the code using Jupyter Notebook (`https://jupyter.org`) because it allows for interactive demonstration for education purposes. In case the reader would like to run Python script, using the following command can convert a Jupyter Notebook file `xxx.ipynb` into a Python script file `xxx.py`:

```
jupyter nbconvert --to script xxx.ipynb
```

**Requirement**. Running our code requires some common packages. We installed Python and most packages through Anaconda. A few other packages might not be installed automatically, such as Pandas, torchvision, and PyTorch, which are required to run our code. Below are the versions of Python and PyTorch used in our work.

- Python version: 3.9.16 [GCC 7.5.0]
- PyTorch version: 2.0.0

We suggest assigning >30GB space to run all the files.

**License**. We release open-source code under the MIT License to foster future research in this field.

**Demo.** The Jupyter notebook files below demonstrate our non-learned method using SAM and $DINOv2_f$. The masked instances generated by SAM are encapsulated by a minimum bounding

---

*Equal contributions; †Corresponding authors.

37th Conference on Neural Information Processing Systems (NeurIPS 2023) Track on Datasets and Benchmarks.

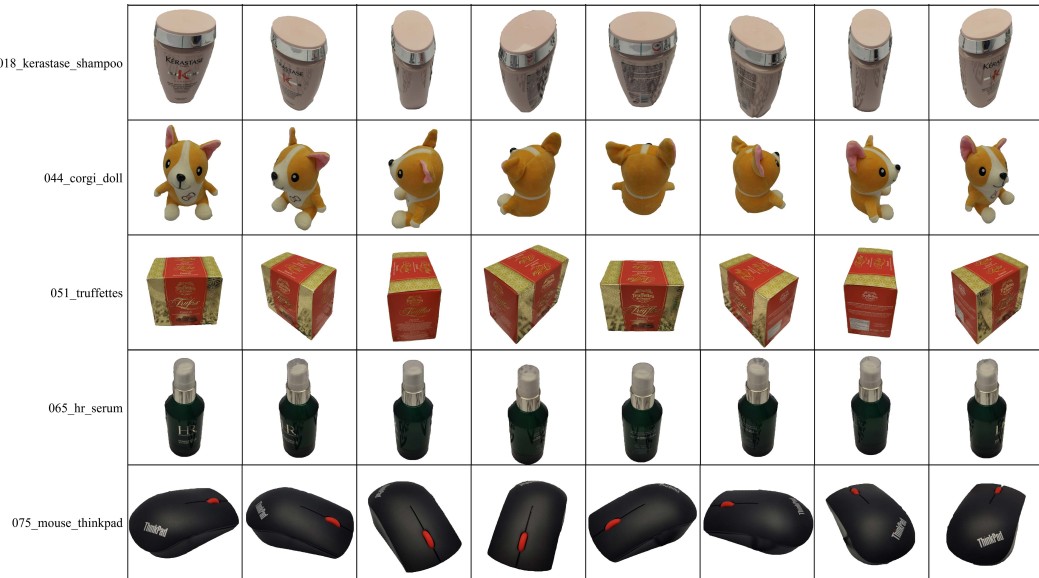

Figure 1: Examples of object instance profile images. We demonstrate profile images of five random object instances (after background removal using GrabCut).

square box and then cropped from the high-resolution testing image. We feed these proposals into DINOv2$_f$ for feature representation, just like how we represent profile images. We run Rank & Select and Stable Matching to determine "well-matched" proposals and instances.

- `demo_get_proposals.ipynb`

  Running this file crops a masked instance from the high-resolution testing image with/without background, and visualizes the cropped proposal regions on the testing image.

- `demo_eval_instance_detection_Stable_Matching.ipynb`

  Running this file extracts features of proposals and object instance profile images by DINOv2$_f$, and implements Stable Matching to return matched proposals and instances.

- `demo_eval_instance_detection_Rank_Select.ipynb`

  Running this file extracts features of proposals and object instance profile images by DINOv2$_f$, and implements Rank & Select to return matched proposals and instances.

## 2 More Visualizations

**Instance profile images**. Fig. 1 presents more visualizations of object instance profile images in our InsDet dataset. The InsDet dataset contains 100 object instances, including sauces, snack foods, office stationery, cosmetics, toiletries, dolls, etc. Each instance is captured at 24 rotation positions (every 15° in azimuth) with 45° elevation view. Profile images are captured at 3072×3072 pixel resolution (some are 3456×3456). We use the GrabCut [2] toolbox to derive foreground masks of instances in profile images. This removes background pixels (such as QR code regions) in the profile images. In practice, we center-crop foreground instances from profile images and downsize the center-crops to 1024×1024. As shown in Fig. 1, we visualize object instances of various shapes and sizes in multiple rotation views.

**Instance proposals produced by SAM**. Fig. 2 shows the segmentation masks produced by SAM on two testing scene images (demonstrated on page 8 of the paper). We downsize the high-resolution testing images to low-resolution (e.g., 768×1024 or 1536×2048) for more efficient and effective segmentation. We observe that SAM can produce high-quality segments, although it also over-segments. Note that at each location seed, SAM produces three proposals which might be object parts or the whole object. We use all of them as proposed detections and let the follow-up step of Stable Matching find the best-matched proposals to instances as the final InsDet results.

**Input proposals to DINOv2$_f$**. As a supplement to Table 5 in the main paper showing whether to remove background in crops, Fig. 3 presents two strategies of feeding proposals to DINOv2$_f$ for

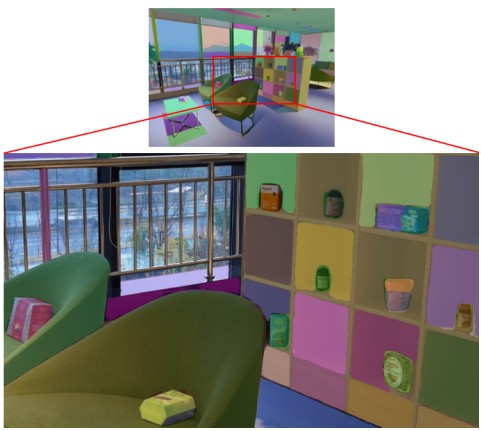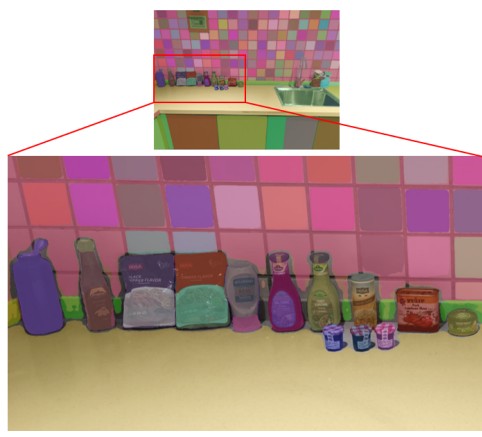

Figure 2: Instance proposals produced by SAM overlaid on the testing images. Note that at each location seed, SAM produces three proposals which might be object parts or the whole object. We use all of them as proposal detections and let the follow-up step of Stable Matching find the best-matched proposals to instances as the final InsDet results.

(a) w/ background                      (b) w/o background

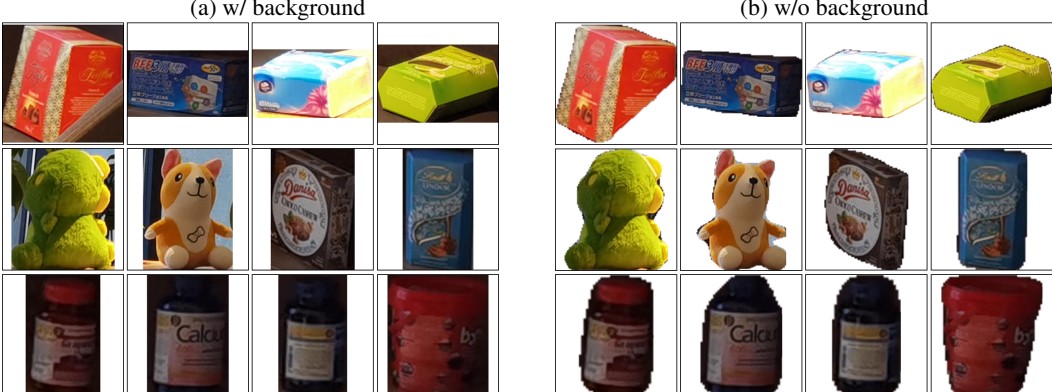

Figure 3: Examples of input proposals to DINOv2$_f$. Based on a proposal given by SAM, we crop the instance using the minimum bounding square ((a) w/ background) or using the segmentation mask ((b) w/o background) before computing the features.

feature computation. One is to crop an instance proposal with its minimum bounding square which keeps the background from the testing image. Another is to use such with background removal (via the mask generated by SAM).

## 3   Ablation Study

We conduct more ablation studies on the traditional cut-paste-learn method. In the cut-paste-learn group of methods, there are four factors influencing the final detection results, i.e., the number of objects inserted per image, the scales of inserted object instances, the blending methods when pasting instances on background images, and the total amount of synthesized training images. We ablate these factors by using the FasterRCNN architecture.

**Number of objects inserted in each background image.** We study InsDet performance with different numbers of objects inserted in each background image in Table 1. We can see that inserting more objects helps train InsDet detectors and achieves better performance. Concretely, FasterRCNN yields 19.54% AP when trained on synthesized training images each of which has 25-35 object instances, better than 17.57% AP when trained on those that have 5-15 object instances per image. But inserting more is not necessarily increasing much further.

**Scales of inserted object instances in synthesizing training images.** We study the impact of the scales of inserted object instances when synthesizing training images in Table 2. The number in square brackets denotes the range of downsampling factors for instance profile images. For example, $[0.1, 0.15]$ denotes that the original instance profile images (256x256 resolution) are randomly scaled

Table 1: **Ablation study: number of objects inserted in each background image**. Basically, inserting more objects helps train InsDet detectors and achieves better performance. Yet, inserting more is not necessarily increasing much further.

| # of objects | AP | AP$_{50}$ | AP$_{75}$ | AP$_s$ | AP$_m$ | AP$_l$ |
|---|---|---|---|---|---|---|
| [5, 15] | 17.57 | 25.98 | 20.49 | 3.55 | 20.31 | 33.50 |
| [15, 25] | 18.20 | 27.76 | 21.09 | 4.45 | 20.66 | 35.51 |
| [25, 35] | 19.39 | 29.14 | **23.09** | 5.03 | 22.04 | 37.73 |
| [35, 45] | **19.60** | **30.30** | 22.82 | **5.44** | **22.32** | **39.17** |

Table 2: **Ablation study: scales of inserted object instances in synthesizing training images**. The scale significantly influences the final detection performance. Inserting objects that are too small (e.g. [0.1, 0.15]) or too large (e.g. [0.5, 1.0]) will not train detectors well. We think this is because the testing images contain more `medium` object instances.

| scale of objects | AP | AP$_{50}$ | AP$_{75}$ | AP$_s$ | AP$_m$ | AP$_l$ |
|---|---|---|---|---|---|---|
| [0.1, 0.15] | 4.72 | 9.63 | 4.16 | 5.48 | 8.93 | 0.72 |
| [0.15, 0.3] | 16.66 | 26.82 | 18.55 | **16.01** | **27.74** | 9.88 |
| [0.15, 0.5] | **19.39** | **29.14** | **23.09** | 5.03 | 22.04 | 37.73 |
| [0.5, 0.8] | 5.43 | 8.16 | 6.08 | 1.79 | 18.72 | **70.20** |
| [0.5, 1.0] | 5.74 | 9.15 | 6.60 | 0.00 | 3.00 | 19.58 |

Table 3: **Ablation study: blending methods between objects and the background images.** We note that: (1) Naive pasting gives the worst performance, since directly pasting objects on background images creates boundary artifacts. (2) Although the other three blending modes do not yield visually perfect results, they could still improve the detection performance. (3) When using all four blending modes when mixing the same background image with the same object displacement, the InsDet performance could be further improved. This is because training on multiple images makes the algorithm less sensitive to these blending factors.

| blending strategy | AP | AP$_{50}$ | AP$_{75}$ | AP$_s$ | AP$_m$ | AP$_l$ |
|---|---|---|---|---|---|---|
| Gaussian | 17.83 | 27.13 | 21.02 | 4.74 | 20.49 | 36.09 |
| motion | 17.92 | 27.57 | 20.85 | 4.69 | 20.76 | 34.78 |
| box blurring | 17.71 | 27.47 | 20.95 | 4.25 | 20.30 | 34.56 |
| naive pasting | 16.53 | 24.77 | 20.17 | 4.25 | 19.07 | 34.86 |
| all | **19.39** | **29.14** | **23.09** | **5.03** | **22.04** | **37.73** |

Table 4: **Ablation study: different amounts of synthesized training images**. Perhaps surprisingly, using 5k synthesized training images is better than training on more images! We conjecture the reasons are that (1) more synthesized images do not bring new signals to help training, (2) domain gaps between synthesized images and real testing images are difficult to close by simply using more such synthetic data, otherwise, training will overfit to them and hence hurt the final InsDet performance.

| # of training images | AP | AP$_{50}$ | AP$_{75}$ | AP$_s$ | AP$_m$ | AP$_l$ |
|---|---|---|---|---|---|---|
| 5k | **19.93** | **30.60** | **23.21** | **5.62** | 22.09 | **38.92** |
| 10k | 19.08 | 29.50 | 21.91 | 4.67 | 21.65 | 37.21 |
| 20k | 19.39 | 29.14 | 23.09 | 5.03 | 22.04 | 37.73 |
| 25k | 19.19 | 29.13 | 22.33 | 4.46 | **22.21** | 36.92 |
| 30k | 18.42 | 28.11 | 21.42 | 4.38 | 21.19 | 36.70 |

by 0.1-0.15 before being pasted on background images. We can see that the scale significantly influences the final detection performance. For example, inserting objects that are too small (e.g. [0.1, 0.15]) or too large (e.g. [0.5, 1.0]) will not train detectors well. We conjecture this is because the testing images contain more *medium* object instances.

**Blending methods between objects and background.** We study four commonly-used blending methods when pasting objects into the background images in Table 3, i.e. Gaussian blurring, motion blurring, box blurring, and naive pasting. We note that: (1) naive pasting yields the worst performance, since this creates boundary artifacts; (2) the other three blending methods work better than naive pasting but do not show significant performance difference; (3) using all the four blending methods together leads to the best performance, significantly better than using any one of them alone.

**Total amount of synthesized training images.** We study InsDet performance by training on different amounts of synthesized images in Table 4. Perhaps surprisingly, using 5k synthesized training images is better than training on more images! We conjecture the reasons are that (1) more

synthesized images do not bring new signals to help training, (2) domain gaps between synthesized images and real testing images are difficult to close by simply using more such synthetic data, otherwise, training will overfit to them and hence hurt the final InsDet performance.

## 4 Datasheet for our Dataset

We follow the datasheet proposed in [1] for documenting our InsDet dataset.

1. Motivation
   (a) For what purpose was the dataset created?
       This dataset was created to study the problem of Instance Detection, i.e., detecting individual object instances from every single-image of cluttered scenes.
   (b) Who created the dataset and on behalf of which entity?
       This dataset was mainly created by Qianqian Shen. Other authors help with logistics.
   (c) Who funded the creation of the dataset?
       [N/A]
   (d) Any other Comments?
       [No]

2. Composition
   (a) What do the instances that comprise the dataset represent?
       RGB images captured by a camera, and annotation files.
   (b) How many instances are there in total?
       There are 2,760 instances including 24*100 profile images (100 objects with each of which having 24 profile images), 200 background images, and 160 testing images.
   (c) Does the dataset contain all possible instances or is it a sample (not necessarily random) of instances from a larger set?
       It contains all possible instances.
   (d) What data does each instance consist of?
       See 2.(a).
   (e) Is there a label or target associated with each instance?
       See 2.(a)
   (f) Is any information missing from individual instances?
       [No]
   (g) Are relationships between individual instances made explicit?
       [Yes] Images and annotations files are associated.
   (h) Are there recommended data splits?
       [Yes] We provide training and testing split for the dataset.
   (i) Are there any errors, sources of noise, or redundancies in the dataset?
       [No] We tried our best to manually annotate bounding boxes of object instances on the testing image, and we did not see visible noise or errors. In profile images of instances, there might be noise in camera pose estimation due to the QR code pasted on the table.
   (j) Is the dataset self-contained, or does it link to or otherwise rely on external resources (e.g., websites, tweets, other datasets)?
       [Yes]
   (k) Does the dataset contain data that might be considered confidential (e.g., data that is protected by legal privilege or by doctor-patient confidentiality, data that includes the content of individuals' non-public communications)?
       [No]
   (l) Does the dataset contain data that, if viewed directly, might be offensive, insulting, threatening, or might otherwise cause anxiety?
       [No]
   (m) Does the dataset relate to people?
       [No]
   (n) Does the dataset identify any subpopulations (e.g., by age, gender)?
       [No]

(o) Is it possible to identify individuals (i.e., one or more natural persons), either directly or indirectly (i.e., in combination with other data) from the dataset?
[No]

(p) Does the dataset contain data that might be considered sensitive in any way (e.g., data that reveals racial or ethnic origins, sexual orientations, religious beliefs, political opinions or union memberships, or locations; financial or health data; biometric or genetic data; forms of government identification, such as social security numbers; criminal history)?
[No]

(q) Any other comments?
[No]

3. Collection Process

(a) How was the data associated with each instance acquired?
For each instance, we capture 24 profile images. For each testing image, we manually annotate bounding boxes on the instances of interest as the ground-truth.

(b) What mechanisms or procedures were used to collect the data (e.g., hardware apparatus or sensor, manual human curation, software program, software API)?
We use a single Leica camera (embedded in a cellphone) to capture profile images and testing scene images. We use GrabCut to obtain foreground masks on the profile images. We manually draw bounding boxes on testing images as the ground-truth.

(c) If the dataset is a sample from a larger set, what was the sampling strategy (e.g., deterministic, probabilistic with specific sampling probabilities)?
[N/A]

(d) Who was involved in the data collection process (e.g., students, crowdworkers, contractors), and how were they compensated (e.g., how much were crowdworkers paid)?
Only authors are involved in the data collection.

(e) Over what timeframe was the data collected?
All images were collected between October 2022 to March 2023.

(f) Were any ethical review processes conducted (e.g., by an institutional review board)?
[No] No ethical review processes were conducted with respect to the collection and annotation of this data.

(g) Does the dataset relate to people?
[No]

(h) Did you collect the data from the individuals in question directly, or obtain it via third parties or other sources (e.g., websites)?
We collected the data by ourselves.

(i) Were the individuals in question notified about the data collection?
[Yes]

(j) Did the individuals in question consent to the collection and use of their data?
[Yes]

(k) If consent was obtained, were the consenting individuals provided with a mechanism to revoke their consent in the future or for certain uses?
[No]

(l) Has an analysis of the potential impact of the dataset and its use on data subjects (e.g., a data protection impact analysis) been conducted?
[N/A]

(m) Any other comments?
[No]

4. Preprocessing, Cleaning and Labeling

(a) Was any preprocessing/cleaning/labeling of the data done (e.g., discretization or bucketing, tokenization, part-of-speech tagging, SIFT feature extraction, removal of instances, processing of missing values)?
[No] The only labeling activity is annotating bounding boxes for instances on testing images. Further annotation cleaning is not needed.

(b) Was the "raw" data saved in addition to the preprocessed/cleaned/labeled data (e.g., to support unanticipated future uses)?
[Yes] We release the raw data on the website of this work.

(c) Is the software used to preprocess/clean/label the instances available?
[Yes] We use open-source software publicly available. We cite them already in the main paper or supplement.

(d) Any other comments?
[No]

5. Uses

(a) Has the dataset been used for any tasks already?
[No]

(b) Is there a repository that links to any or all papers or systems that use the dataset?
[No] We will add future papers that use this dataset in the website of this work.

(c) What (other) tasks could the dataset be used for?
The dataset can also be used to study small object detection, 3D reconstruction from sparse views, real-time object detection, etc.

(d) Is there anything about the composition of the dataset or the way it was collected and preprocessed/cleaned/labeled that might impact future uses?
[Yes] Capturing high-resolution images is cheap but requires more computation resources in subsequent tasks. This might impact future work related to how to capture high-resolution images, how to trade off performance and resolution, etc.

(e) Are there tasks for which the dataset should not be used?
The usage of this dataset should be limited to the scope of object instance detection.

(f) Any other comments?
[No]

6. Distribution

(a) Will the dataset be distributed to third parties outside of the entity (e.g., company, institution, organization) on behalf of which the dataset was created?
[Yes] We expect other websites re-distribute our dataset.

(b) How will the dataset be distributed (e.g., tarball on website, API, GitHub)?
The dataset could be accessed on a GitHub webpage.

(c) When will the dataset be distributed?
The dataset will be released to the public upon acceptance of this paper. We provide some demo data and visualizations for the review process.

(d) Will the dataset be distributed under a copyright or other intellectual property (IP) license, and/or under applicable terms of use (ToU)?
We release our benchmark under MIT license.

(e) Have any third parties imposed IP-based or other restrictions on the data associated with the instances?
[No]

(f) Do any export controls or other regulatory restrictions apply to the dataset or to individual instances?
[No]

(g) Any other comments?
[No]

7. Maintenance

(a) Who is supporting/hosting/maintaining the dataset?
Qianqian Shen will be responsible for maintaining the dataset.

(b) How can the owner/curator/manager of the dataset be contacted (e.g., email address)?
E-mail addresses are at the top of the paper.

(c) Is there an erratum?
[No] When errors are discerned, we will announce erratum on our website.

(d) Will the dataset be updated (e.g., to correct labeling errors, add new instances, delete instances')?
[Yes] We hope to expand this dataset with more instances and testing images.

(e) If the dataset relates to people, are there applicable limits on the retention of the data associated with the instances (e.g., were individuals in question told that their data would be retained for a fixed period of time and then deleted)?
[N/A]

(f) Will older versions of the dataset continue to be supported/hosted/maintained?
[Yes]

(g) If others want to extend/augment/build on/contribute to the dataset, is there a mechanism for them to do so?
[No]

(h) Any other comments?
[No]