# OpenReview forum: "A High-Resolution Dataset for Instance Detection with Multi-View Object Capture"
_NeurIPS.cc/2023/Track/Datasets_and_Benchmarks — NeurIPS 2023 Datasets and Benchmarks Poster_

### Official Review · Reviewer_f4oD · 2023-06-29
**Clean, high-resolution dataset, but potentially redundant**

**Rating:** 6
**Confidence:** 3
**Correctness:** Yes.
**Clarity:** The paper is easy to read and easy to…

**Strengths:**

1. The dataset presented in the paper is clean, and it provides a dataset for a task that lacks many relevant and sufficiently large datasets.
2. The images in the dataset are clearly of high-resolution, which is important to the usefulness of the dataset.
3. The dataset is tailored SPECIFICALLY to instance detection, which is useful for the convenient development of instance detection models and methods.

**Additional Feedback:**

None.

**Documentation:**

Yes.

**Ethics:**

No.

**Limitations:**

While the limitations listed in the paper do not seem to mention societal impact, I agree that this work does not have any significant negative societal impact.

**Opportunities For Improvement:**

1. Although there is definitely a diversity of backgrounds in the images, it may be a little limited in that they all seem to be indoor and of the same general "type".
2. There should be more of a discussion on the significance of the models' performance on the dataset. So what if some models perform better than others on this dataset? Does this speak to the merits of the dataset, remembering that this is a datasets & benchmarks paper?
3. My biggest comment is: Please explain why COCO or other large object detection datasets cannot be used for the task of instance detection. I am happy to raise my rating if this is sufficiently explained.

**Relation To Prior Work:**

Yes.

**Summary And Contributions:**

Instance detection is a less focused-on method than object detection, partly because of a lack of large, useful datasets for the task. The dataset contains 100 distinct object instances. A model trained on this dataset would be trained to identify instances of these objects while ignoring other objects. Even though the images themselves are high-resolution, the objects are small, meaning that they only take up a small portion of each image. This makes instance detection on this dataset a more difficult task. The authors show that stronger instance detection models perform better on the dataset. Furthermore, in general, all models seem to perform sub-optimally on the dataset, which shows that the dataset is a difficult and useful benchmark.

---

> ### Author Response · Authors · 2023-08-17
> **thanks and responses**
>
> We thank Reviewer f4oD for the valuable comments. Reviewer f4oD finds our dataset to be "clean", "clearly of high-resolution which is important to the usefulness of the dataset", etc. Reviewer f4oD has three major concerns and we address them below.
>
>
> > **Reviewer f4oD thinks the images in our dataset are "a little limited in that they all seem to be indoor and of the same general type".**
>
> Using indoor scenes is a typical thing in the literature of Instance Detection (InsDet), as done in previous InsDet datasets such as GMU [15], AVD [1], and Grocery [4]. This is primarily because InsDet is better explored in indoor scenes motivated by indoor assistive robots and micro-fulfillment robots in retail that need to pick items from boxes or shelves (Line23). Importantly, our dataset captures testing data from 14 distinct scenes, whereas prior datasets have less than 10 indoor scenes (cf. Table 1 in our paper). In this context and the field of InsDet, our dataset is more general than existing ones instead of "limited".
>
>
> > **Reviewer f4oD suggests discussing more "on the significance of the models' performance on the dataset".**
>
> Good point! Our paper contains the precision-recall curves as Fig. 4, showing detailed precision and recall performance at different operating points (i.e., threshold of detection confidence scores). We can see the non-learned method SAM+DINOv2 achieves higher precision and higher recall than others at almost all the operating points. Furthermore, we have studied average recall (AR) of the compared methods, see Table 3 in the revised paper (also copied below for your convenience). AR measures recall performance by ignoring classification accuracy. "AR @ max10" means AR within the top-10 ranked detections. In computing AR, we rank detections by using the detection confidence scores of the learning-based methods (e.g., FasterRCNN) or similarity scores in the non-learned methods (e.g., SAM+DINO$_f$). AR$_s$, AR$_m$ and AR$_l$ are breakdown of AR for small, medium and large testing object instances. From both precision-recall curves and AR results, we can see that (1) detectors with stronger architectures perform better w.r.t all these metrics, (2) the non-learned methods that use SAM generally recall more instances than the end-to-end models (e.g.,FasterRCNN), and (3) all methods suffer from detecting small instances.
>
>   | metric | FasterRCNN | RetinaNet | CenterNet | FCOS | DINO | SAM+DINO$_f$ | SAM+DINOv2$_f$ |
>   | --- | --- | ---- | --- | --- | --- | --- | --- |
>    AR @ max10      | 26.24 | 26.33 | 23.55 | 25.82 | 29.84 | 31.25 | 40.02
>    AR @ max100    | 39.24 | 49.38 | 44.72 | 46.28 | 54.22 | 63.05 | 63.06
>    AR$_s$ @ max100  | 14.83 | 22.04 | 17.84 | 22.09 | 32.00 | 31.65 | 31.11
>    AR$_m$ @ max100 | 44.87 | 56.76 | 52.03 | 52.85 | 59.43 | 70.01 | 70.40
>    AR$_l$ @ max100   | 60.05 | 69.69 | 64.58 | 64.11 | 72.92 | 90.63 | 90.36
>
>
> > **Reviewer f4oD asks "why COCO or other large object detection datasets cannot be used for the task of instance detection".**
>
> COCO and other object detection datasets are designed for general object detection, which requires detecting all objects belonging to some predefined classes, e.g., “cup”. For example, the class “cup” may have various object instances of “coffee cup”, “tea cup”, “paper cup”, “thermal mug” and so on. In contrast, instance detection (InsDet) requires detecting particular object instances predefined by some visual examples. Loosely speaking, InsDet treats each predefined object instance as a single class. For example, in InsDet, two thermal mugs that are different only in color are treated as two different classes. In sum, COCO and other object detection datasets do not contain instance labels to support the exploration of InsDet. We have added an illustrative figure (Fig. 1) in the revised paper, hoping this figure will help readers better understand the difference between general object detection and InsDet.

---

> > ### Comment · Reviewer_f4oD · 2023-08-29
> > **Acknowledgment of Merits**
> >
> > Thank you for your detailed responses! I do especially agree with your last point about this dataset being more fine-grained than COCO. I therefore raise my rating.

---

> > > ### Author Response · Authors · 2023-08-30
> > >
> > > We thank Reviewer f4oD for letting us know that our responses have effectively addressed the reviewer's concerns, especially the last point which is the reviewer's "biggest comment". While our responses have convinced the reviewer to upgrade the rating (to "6: Marginally above acceptance threshold"), we are delighted to address existing or any other concerns which might prevent upgrading the rating further!

---

### Official Review · Reviewer_Yusx · 2023-07-20
**Some confusion on the main selling point**

**Rating:** 7
**Confidence:** 3
**Correctness:** The evaluation protocol seems fair.
**Clarity:** The paper is well-written.

**Strengths:**

Main strengths:

 - the paper is well-written and easy to follow;
 - the proposed dataset clearly represents an improvement w.r.t. previous solutions for studying the InsDet task. Moreover, it is well described in all its parts, from data collection tools, data description, included categories, segmentation, and labeling approaches;
 - the standardization effort for improving the InsDet study field is commendable;
 - the proposed non-learned InsDet strategy is interesting and obtains very good results.

**Additional Feedback:**

Some randomly ordered points:

 - the introduction is rigidly structured in paragraphs, but it relies on an image caption to introduce the dataset;
 - the dataset from [22] is not included in Table 1;
 - the Instance Detection paragraph in the Related Works section should come before the Object Detection one;
 - the split of test images in easy and hard seems a bit arbitrary;
 - in section 3.2, what does it mean that the 100 instances come from 14 indoor scenes? Is this the number of test scenes?
 - it is not clear why the QR codes are inserted in the profile images and then a manual segmentation tool (grabcut) is used to remove them;
 - in section 4.2, what does it mean that SAM has a high recall on the proposed dataset? Would it be possible to provide an empirical quantitative measure of this phenomenon?
 - in the implementation details section the data synthesis protocol should be presented before the object detectors training protocol. With the current order, it is not clear which validation data is used for model selection when reading the training protocol part.

**Documentation:**

The description of the dataset is exhaustive.

**Ethics:**

I do not see ethical concerns.

**Limitations:**

The authors addressed the limitations and societal impact.

**Opportunities For Improvement:**

There are some significant opportunities for improvement.

**Main paper message**

The main purpose of this paper is the presentation of a novel dataset and related evaluation benchmark for the instance detection task. This point, however, is often overshadowed by the presentation, description, and discussion of the novel non-learned InsDet method. Indeed, while the introduction of this novel InsDet paradigm is clearly an interesting element, it also generates a bit of confusion on which selling point the authors are more focused on. For example, Section 4 provides a lot of information on this novel paradigm, reserving only a very small paragraph for the traditional cut-paste-learn one. Moreover, section 5.2 which should provide additional empirical analyses is again completely focused on this novel method, and thus no additional analyses are provided for standard methods or dataset-related aspects.

For example, it is probable that the data synthesis protocol used for the definition of training and validation data for detectors based on the cut-paste-learn paradigm highly influences the performance of these detectors. However, the authors do not perform an analysis of this. In practice a single strategy for data synthesis is provided and then the obtained dataset is used as-is. In the limitation sections the authors also argue that using geometric cues for image synthesis could improve the results. This certainly would be an interesting experiment, but an even simpler analysis could be performed by changing the number of objects inserted in each background image, their relative size, the total size of generated training samples, and so on. For example, it is not clear which are the differences among the various blending options and how each option contributes to the final result.

**A bit of confusion on the related works**

From the point of view of a reader who is not an expert in the Instance Detection task and literature, the abstract, introduction, and related work sections present some confusing parts. The abstract and introduction clearly mention that this paper wants to propose a novel setup and protocol for InsDet. Still, it is not clear at all which are the disadvantages of the previous protocols, apart from the fact that they are small. Only further reading allows understanding that the main problem of the literature till this point is that it completely misses a standard evaluation protocol.
The presentation of the existing InsDet literature does not allow to build a clear picture of the situation in terms of which are the categories of existing approaches. The cut-paste-learn framework is the only category which is analyzed more deeply.
Moreover, both the introduction and the related works section (in the InsDet paragraph) reference a number of papers which are not really focusing on the InsDet task, but the reader can understand this only by opening the referenced papers. For example from my understanding [26] and [17] are not methods for InsDet, even if they are presented as such and particularly as belonging to the cut-paste-learn group.

**Realism of the proposed protocol**

Multiple times and particularly at the beginning of section 3.1 the authors argue that they followed a real-world application's requirements in the design of their dataset and protocol. This application is the design of assistive robots. The realism and practicality of the proposed paradigm are however quite subjective. For example, I do not find particularly realistic the fact that profile images have to be captured with instances laying on a plane on which QR codes are printed(I would expect that for enabling customization of a robot knowledge the profile images should be captured when the object instances are handheld by a human, like in the COSDA-HR dataset [A]), nor the fact that it is necessary to collect *empty* background images in order to synthesize training data. From this point of view, the novel non-learned InsDet approach seems to be more practical as it does not use these background images.

[A] Ikki et al, "Object Recognition With Continual Open Set Domain Adaptation for Home Robot", WACV 2021

**Relation To Prior Work:**

As pointed out in the "Opportunities for improvement", the relation to prior work could be clarified better.

**Summary And Contributions:**

This paper focuses on the Instance Detection task and proposes both a novel large-scale dataset and a benchmark built on top of it. On this evaluation bed, the authors test the performance of a standard learning-based InsDet method and a novel non-learned approach, which obtains state-of-the-art results.

Main contributions:

 - a novel dataset for InsDet. The dataset has a larger number of instances w.r.t. previous public datasets and uses high-resolution images;
 - an evaluation benchmark based on the new dataset and designed to standardize the protocol for evaluating InsDet models. On this benchmark, the authors evaluate detectors based on the standard cut-paste-learn strategy;
 - a novel non-learned method for InsDet which exploits only pretrained models and obtains sota results.

---

> ### Author Response · Authors · 2023-08-17
> **thanks and responses [part-1]**
>
> We thank Reviewer Yusx for the insightful comments and having "no reservation" about the paper quality, e.g., "the paper is well-written and easy to follow", "it is well described in all its parts'',  "standardization effort is commendable'', and "the proposed non-learned InsDet strategy is interesting and obtains very good results". We respond to your major comments below, and have happily revised our paper according to your valuable suggestions.
>
> > **Reviewer Yusx finds the current paper organization is imbalanced in that it spends much space on the novel non-learned method but insufficiently describes traditional cut-paste-learn methods.**
>
> Great comment! We have revised our manuscript by (1) moving some ablation studies of our proposed non-learned methods (e.g., SAM+DINOv2) to the supplement, and (2) including results w.r.t average recall (AR) of both non-learned and learning-based methods.
> By the way, we have included comprehensive ablation studies for the cut-paste-learn methods in the revised supplement (see details in part-2 of our response.
>
>
> > **Reviewer Yusx asks what it means that "the 100 instances come from 14 indoor scenes".**
>
> Apologies for the confusion. It means that testing images are captured in 14 indoor scenes where there are 100 object instances defined for Instance Detection. We have rewritten this sentence in the revised paper (Line143).
>
> > **Reviewer Yusx asks why QR codes are inserted in the profile images and why using GrabCut to segment the foreground instances again.**
>
> Using QR code is a typical thing in previous related datasets (e.g., LM [21], and LM-O [5], and RU-APC [38]), intending to offer an opportunity to estimate object/camera pose for 3D reconstruction, which has a potential to help Instance Detection. We follow the literature for this purpose too. Yet, segmenting the foreground instances is to paste them on background images, allowing cut-paste-learn methods to train their detectors.
>
> > **Reviewer Yusx asks if it would be possible to provide an empirical quantitative measure that shows SAM has a high recall on the proposed dataset.**
>
> Great point! The table below lists results of average recall (AR) by all the methods studied in our paper (we added this table as Table 3 in our revised paper). As for the metric, "AR @ max10" means AR within the top-10 ranked detections. In computing AR, we rank detections by using the detection confidence scores of the learning-based methods (e.g., FasterRCNN) or similarity scores in the non-learned methods (e.g., SAM+DINO$_f$). AR$_s$, AR$_m$ and AR$_l$ are breakdown of ARs for "small", "medium" and "large" testing object instances. Results show that (1) the non-learned methods that use SAM generally recall more instances than others, and (2) all methods suffer from small instances. In sum, results show that methods yielding higher AR also achieve higher AP(cf. Table 2 in the paper).
>
>   | metric | FasterRCNN | RetinaNet | CenterNet | FCOS | DINO | SAM+DINO$_f$ | SAM+DINOv2$_f$ |
>   | --- | --- | ---- | --- | --- | --- | --- | --- |
>    AR @ max10      | 26.24 | 26.33 | 23.55 | 25.82 | 29.84 | 31.25 | 40.02
>    AR @ max100    | 39.24 | 49.38 | 44.72 | 46.28 | 54.22 | 63.05 | 63.06
>    AR$_s$ @ max100  | 14.83 | 22.04 | 17.84 | 22.09 | 32.00 | 31.65 | 31.11
>    AR$_m$ @ max100 | 44.87 | 56.76 | 52.03 | 52.85 | 59.43 | 70.01 | 70.40
>    AR$_l$ @ max100   | 60.05 | 69.69 | 64.58 | 64.11 | 72.92 | 90.63 | 90.36

---

> > ### Author Response · Authors · 2023-08-17
> > **thanks and responses [part-2]**
> >
> > > **Reviewer Yusx suggests ablation studies for cut-paste-learn methods because their performance is highly influenced by multiple factors in training data synthesis, including the number of inserted objects in each background image, their relative size, the number of generated training images, and blending methods.**
> >
> > Great suggestion! In fact, we did tune these factors for cut-paste-learn methods (through training the FasterRCNN detector for InsDet) and report numbers in the paper with the best tuned choices. Apologies for not including them in our initial submission. Because the ablation studies are too detailed and take too much space, we put them in the revised supplement (Table S3-S7  therein); we are delighted to include any of the tables in the main paper if the reviewer thinks necessary. Below we copy the tables for your convenience.
> >
> >  - The table below studies InsDet performance with different numbers of objects inserted in each background image. We can see that inserting more objects helps train InsDet detectors that achieves better performance. Concretely, FasterRCNN yields 19.54 AP when trained on synthesized training images each of which has 25-35 object instances, better than 17.57% AP when trained on those that have 5-15 object instances per image. But inserting more is not necessarily increasing much further.
> >
> >    |# of objects| AP | AP50 | AP75 | AP$_s$ | AP$_m$ | AP$_l$ |
> >    | ------- | --------- | -------- | --------- | --------- | --------- | --------- |
> >    [5, 15] | 17.57 | 25.98 | 20.49 | 3.55 | 20.31 | 33.50
> >    [15, 25] | 18.20 | 27.76 | 21.09 | 4.45 | 20.66 | 35.51
> >    [25, 35] | 19.39 | 29.14 | 23.09 | 5.03 | 22.04 | 37.73
> >    [35, 45] | 19.60 | 30.30 | 22.82 | 5.44 | 22.32 | 39.17
> >
> > - The table below studies the impact of the scales of inserted object instances when synthesizing training images. The number in square brackets denotes the range of downsampling factors for instance profile images. For example, $[0.1, 0.15]$ denotes that the original instance profile images (256x256 resolution) are randomly scaled by 0.1-0.15 before pasted on background images. We can see that the scale significantly influences the final detection performance. For example, inserting objects that are too small (e.g. [0.1, 0.15]) or too large (e.g. [0.5, 1.0]) will not train detectors well. We think this is because the testing images contain more "medium" object instances.
> >
> >    |scale of objects| AP | AP50 | AP75 | AP$_s$ | AP$_m$ | AP$_l$ |
> >    | ------- | --------- | --------- | --------- | -------- | --------- |--------- |
> >    [0.1, 0.15] | 4.72   | 9.63   | 4.16   | 5.48   | 8.93   | 0.72
> >    [0.15, 0.3] | 16.66 | 26.82 | 18.55 | 16.01 | 27.74 | 9.88
> >    [0.15, 0.5] | 19.39 | 29.14 | 23.09 | 5.03 | 22.04 | 37.73
> >    [0.5, 0,8]   | 5.43   | 8.16   | 6.08   | 1.79   | 18.72 | 70.20
> >    [0.5, 1.0]   | 5.74   | 9.15   | 6.60   | 0.00   | 3.00   | 19.58
> >
> > - Below studies four commonly-used blending methods when pasting instances on background images:  Gaussian blurring, motion blurring , box blurring, and naive pasting.  Results show that: (1) naive pasting yields the worst performance, since this creates boundary artifacts; (2) the other three blending methods work better than naive pasting but do not show significant performance difference; (3) using all the four blending methods leads to the best performance, significantly better than using any one of them alone.
> >
> >    |blending mode| AP | AP50 | AP75 | AP$_s$ | AP$_m$ | AP$_l$ |
> >    | ------- | --------- | --------- | --------- | -------- | --------- |--------- |
> >    Gaussian        | 17.83 | 27.13 | 21.02 | 4.74 | 20.49 | 36.09
> >    motion            | 17.92 | 27.57 | 20.85 | 4.69 | 20.76 | 34.78
> >    box blurring    | 17.71 | 27.47 | 20.93 | 4.25 | 20.30 | 34.56
> >    naive pasting   | 16.53 | 24.77 | 20.17 | 4.25 | 19.07 | 34.86
> >    all                   | 19.39 | 29.14 | 23.09 | 5.03 | 22.04 | 37.73
> >
> > - Below studies InsDet performance by training on different amounts of synthesized images. Perhaps surprisingly, using 5k synthesized training images is better than training on more images! We conjecture the reasons are that (1) more synthesized images do not bring new signals to help training, (2) domain gaps between synthesized images and real testing images are difficult to close by simply using more such synthetic data, otherwise training will overfit to them and hence hurt the final InsDet performance.
> >
> >    |# of training images| AP | AP50 | AP75 | AP$_s$ | AP$_m$ | AP$_l$ |
> >    | ------- | --------- | --------- | --------- | -------- | --------- |--------- |
> >    5k   | 19.93 | 30.60 | 23.21 | 5.62 | 22.09 | 38.92
> >    10k | 19.08 | 29.50 | 21.91 | 4.67 | 21.65 | 37.21
> >    20k | 19.39 | 29.14 | 23.09 | 5.03 | 22.04 | 37.73
> >    25k | 19.19 | 29.13 | 22.33 | 4.46 | 22.21 | 36.92
> >    30k | 18.42 | 28.11 | 21.42 | 4.38 | 21.19 | 36.70

---

> ### Comment · Reviewer_Yusx · 2023-08-25
> **Final rate**
>
> The authors have addressed most of my concerns, I have decided to raise my rating to 7

---

> > ### Author Response · Authors · 2023-08-25
> > **thank you for the comment**
> >
> > We thank Reviewer Yusx for letting us know that our responses have addressed most of the reviewer's concerns, leading to an upgraded rating of "7: Good paper, accept". We highly appreciate the constructive feedback provided by the reviewer. Moreover, we are delighted to address any other concerns which might prevent upgrading the rating further. Further advice on improving our submission is also very welcome!

---

### Official Review · Reviewer_3YDD · 2023-07-22

**Rating:** 7
**Confidence:** 3
**Correctness:** The dataset is constructed in a sound…
**Clarity:** The writting of this paper is good an…

**Strengths:**

1. This paper is well-written and easy to follow.
2. The authors discussed the shortcomings of previous InsDet-related work and clearly articulated their motivation for proposing a new dataset.
3. Compared to previous datasets, the dataset proposed by the authors has better objects and scenes, as well as a higher pixel resolution.
4. The authors provided two simple and robust baselines for Instance Detection.

**Additional Feedback:**

Please refer to Opportunities For Improvement.

**Documentation:**

The authors propose the details of data collection and organization. They offer a subset of this dataset and codes for how to use this dataset in the submission.

**Ethics:**

I consider there are not ehical concerns in this submission.

**Limitations:**

The authors believe that directly applying the dataset they propose to the real world could be risky, and this work has not considered practical applications, such as cheaper hardware and faster processing methods. The authors have analyzed the limitations of their work and potential negative societal impact, which have not been resolved in the article. The authors consider these as future work.

**Opportunities For Improvement:**

1. Most people might not fully understand the difference between Instance Detection and General Object Detection. The authors could provide a more detailed introduction to what the task of Instance Detection entails at the beginning of the article. Also, they could compare their proposed dataset with COCO. For example, in the Instance Detection dataset, an object is considered a class.
2. The author adopt easy and hard as condition in the evaluation. The tags "easy" and "hard" are differentiated based on the level of clutter and occlusion. However, in the field of 2D Detection, factors affecting the detection performance include not only occlusion but also the size of the objects. I suggest that the authors present the distribution of object sizes in the dataset and differentiate in the evaluation based on the size of the objects. This could help readers understand the performance differences of different detectors or methods at different scales.
3. Providing the results of AR (Average Recall) of proposals (ignoring the correctness of classification) could help readers understand whether the differences between the methods are due to instance classification or detection.

**Relation To Prior Work:**

The author discuss the difference between the proposed dataset and other prior datasets. They discuss the motivation of proposing a new dataset for Instance Detection.

**Summary And Contributions:**

This paper proposes a new dataset for Instance Detection. The dataset contains high-resolutions multi-view images of 100 distinct object instances from 14 indoor scenes. Then the authors propose (1) a simple but strong baseline based on Cut-Paste-Learn and (2) a simple, non-learned method based on proposal generation and matching. They conduct these methods on the proposed datasets and comparing the performances of these methods and share interesting findings.

---

> ### Author Response · Authors · 2023-08-17
> **thanks and response to the reviews**
>
> We thank Reviewer 3YDD for the insightful comments. Reviewer 3YDD finds our paper "well-written and easy to follow" that "clearly articulated their motivation for proposing a new dataset", and the "two simple and robust baselines". Reviewer 3YDD's main concerns are exactly the three constructive suggestions which we respond to in the global message. Therefore, our response below is roughly a copy-paste of our global response.
>
> > **Reviewer 3YDD suggests "providing a more detailed introduction" to help readers to "fully understand the difference between Instance Detection and general Object Detection".**
>
> Great suggestion! We have added an illustrative figure (Fig. 1 on the 2nd page in the revised paper) and revised Line25-27, introducing their difference. We believe the revision can help readers understand the problem of InsDet better.
>
> > **Reviewer 3YDD suggests breakdown analysis with respect to object sizes other than using "easy" and "hard" tags for testing scenes.**
>
> Great advice! We plot the distribution of object sizes (measured by bounding box area) for the testing set, and define “small”, “medium” and “large” objects as in the following table. This follows the spirit of the COCO dataset (a benchmark for general object detection) which tags objects by "small", "medium", and "large". Please refer to our updated supplement for the distribution and splits (Fig. S4 and Table S1 therein).
>
> |size | bounding box area |
> | --- | --- |
> small | <200$^2$
> medium | 200$^2$ - 400$^2$
> large | > 400$^2$
>
> We have included a breakdown analysis as Table 2 in our revised paper. We copy the table below for the convenience of rebuttal, where AP$_s$, AP$_m$ and AP$_l$ are AP metrics for "small", "medium", and "large" testing instances. Results show that all methods perform significantly worse on "small" than "large", suggesting future work of improving small instance detection. Moreover, our conclusions still hold that (1) detectors with stronger architecture perform better, i.e. SAM+DINOv2$_f$ (14.58 AP$_s$) vs. FasterRCNN (5.03 AP$_s$), (2) the non-learned method (SAM+DINO$_f$ and SAM+DINOv2$_f$ outperform end-to-end trained models (e.g., FasterRCNN, FCOS, etc.), and (3) all the methods perform much more poorly on hard instances (which are either small or from more cluttered scenes).
>
> | Method | AP | AP50 | AP75 | AP$_s$ | AP$_m$ | AP$_l$ |
> | --- | --- | --- | --- | ---- | --- |--- |
> FasterRCNN           | 19.54 | 29.21 | 23.26 | 5.03 | 22.20 | 37.97 |
> RetinaNet               | 22.22 | 31.19 | 24.98 | 5.48 | 25.80 | 42.71 |
> CenterNet               | 21.12 | 32.72 | 23.60 | 5.90 | 24.15 | 40.38 |
> FCOS                     | 22.11 | 32.43 | 25.42 | 6.17 | 26.46 | 38.13 |
> DINO                      | 28.09 | 39.73 | 32.29 | 11.51 | 31.60 | 48.35|
> SAM+DINO$_f$     | 36.97 | 44.13 | 40.42 | 11.93 | 40.85 | 62.67 |
> SAM+DINOv2$_f$ | 41.61 | 49.10 | 45.95 | 14.58 | 45.83 | 69.14 |
>
>
> > **Reviewer 3YDD suggests providing the results of AR (Average Recall) of proposals by ignoring the correctness of classification to "help readers understand whether the differences between the methods are due to instance classification or detection."**
>
> Great point! The table below lists results of AR by all the methods studied in our paper (we added this table as Table 3 in our revised paper). As for the metric, "AR @ max10" means AR within the top-10 ranked detections. In computing AR, we rank detections by using the detection confidence scores of the learning-based methods (e.g., FasterRCNN) or similarity scores in the non-learned methods (e.g., SAM+DINO$_f$). AR$_s$, AR$_m$ and AR$_l$ are breakdown of ARs for "small", "medium" and "large" testing object instances. Results show that (1) the non-learned methods that use SAM generally recall more instances than others, and (2) all methods suffer from small instances. In sum, results show that methods yielding higher recall achieve higher AP metrics (cf. Table 2 in the paper).
>
>   | metric | FasterRCNN | RetinaNet | CenterNet | FCOS | DINO | SAM+DINO$_f$ | SAM+DINOv2$_f$ |
>   | --- | --- | ---- | --- | --- | --- | --- | --- |
>    AR @ max10      | 26.24 | 26.33 | 23.55 | 25.82 | 29.84 | 31.25 | 40.02
>    AR @ max100    | 39.24 | 49.38 | 44.72 | 46.28 | 54.22 | 63.05 | 63.06
>    AR$_s$ @ max100  | 14.83 | 22.04 | 17.84 | 22.09 | 32.00 | 31.65 | 31.11
>    AR$_m$ @ max100 | 44.87 | 56.76 | 52.03 | 52.85 | 59.43 | 70.01 | 70.40
>    AR$_l$ @ max100   | 60.05 | 69.69 | 64.58 | 64.11 | 72.92 | 90.63 | 90.36

---

> > ### Comment · Reviewer_3YDD · 2023-08-29
> >
> > Thanks for your response. Most of my concerns have been addressed.

---

> > > ### Author Response · Authors · 2023-08-29
> > > **thank you for the final comment**
> > >
> > > We thank Reviewer 3YDD for letting us know that our responses have addressed most of the reviewer's concerns, leading to an upgraded rating of "7: Good paper, accept". We highly appreciate the constructive feedback provided by the reviewer, which clearly improves the quality of our paper!

---

### Author Response · Authors · 2023-08-17
**Summary of reviews, rebuttal, and revisions**

Dear Area Chairs and Reviewers,

We thank all of you for the hard work of reviewing our paper and valuable comments. Our submission introduces a new dataset to foster the research of Instance Detection. All reviewers think our dataset is clean, useful and important, and appreciate our standardization effort, detailed documentation, and the novel simple non-learned methods. As a result, reviewers give positive ratings (two 6-rating, and one 5-rating). During this author-reviewer discussion period, we provide a general message below to answer some common questions, and separate responses to individual reviewers. We have also updated our revised paper and supplement, wherein we mark major revisions in blue.

> **All the reviewers either hint or constructively suggest introducing the problem of Instance Detection (InsDet) by contrasting the problem of general Object Detection (ObjDet) to help readers understand InsDet better.**

Great suggestion! We have added an illustrative figure (Fig. 1 on the 2nd page in the revised paper) and revised Line25-27, introducing their difference. We believe the revision can help readers understand the problem of InsDet better.

>**All reviewers either hint or constructively ask for evaluating models w.r.t object instance size, other than using "easy" and "hard" tags for testing scenes.**

Great advice! We plot the distribution of object sizes (measured by bounding box area) for the testing set, and define “small”, “medium” and “large” objects as in the following table. This follows the spirit of the COCO dataset (a benchmark for general object detection) which tags objects by "small", "medium", and "large". Please refer to our updated supplement for the distribution and splits (Fig. S4 and Table S1 therein).

|size | bounding box area |
| --- | --- |
small | <200$^2$
medium | 200$^2$ - 400$^2$
large | > 400$^2$

We have included a breakdown analysis as Table 2 in our revised paper. We copy the table below for the convenience of rebuttal, where AP$_s$, AP$_m$ and AP$_l$ are AP metrics for "small", "medium", and "large" testing instances. Results show that all methods perform significantly worse on "small" than "large", suggesting future work of improving small instance detection. Moreover, our conclusions still hold that (1) detectors with stronger architecture perform better, i.e. SAM+DINOv2$_f$ (14.58 AP$_s$) vs. FasterRCNN (5.03 AP$_s$), (2) the non-learned method (SAM+DINO$_f$ and SAM+DINOv2$_f$ outperform end-to-end trained models (e.g., FasterRCNN, FCOS, etc.), and (3) all the methods perform much more poorly on hard instances (which are either small or from more cluttered scenes).

| Method | AP | AP50 | AP75 | AP$_s$ | AP$_m$ | AP$_l$ |
| --- | --- | --- | --- | ---- | --- |--- |
FasterRCNN           | 19.54 | 29.21 | 23.26 | 5.03 | 22.20 | 37.97 |
RetinaNet               | 22.22 | 31.19 | 24.98 | 5.48 | 25.80 | 42.71 |
CenterNet               | 21.12 | 32.72 | 23.60 | 5.90 | 24.15 | 40.38 |
FCOS                     | 22.11 | 32.43 | 25.42 | 6.17 | 26.46 | 38.13 |
DINO                      | 28.09 | 39.73 | 32.29 | 11.51 | 31.60 | 48.35|
SAM+DINO$_f$     | 36.97 | 44.13 | 40.42 | 11.93 | 40.85 | 62.67 |
SAM+DINOv2$_f$ | 41.61 | 49.10 | 45.95 | 14.58 | 45.83 | 69.14 |

>**All reviewers either hint or constructively ask for evaluating models w.r.t average recall (AR) by ignoring classification accuracy. This helps readers understand whether models achieving higher AR yields higher AP.**

Great point! The table below lists results of AR by all the methods studied in our paper (we added this table as Table 3 in our revised paper). As for the metric, "AR @ max10" means AR within the top-10 ranked detections. In computing AR, we rank detections by using the detection confidence scores of the learning-based methods (e.g., FasterRCNN) or similarity scores in the non-learned methods (e.g., SAM+DINO$_f$). AR$_s$, AR$_m$ and AR$_l$ are breakdown of ARs for "small", "medium" and "large" testing object instances. Results show that (1) the non-learned methods that use SAM generally recall more instances than others, and (2) all methods suffer from small instances. In sum, results show that methods yielding higher recall achieve higher AP metrics (cf. Table 2 in the paper).

  | metric | FasterRCNN | RetinaNet | CenterNet | FCOS | DINO | SAM+DINO$_f$ | SAM+DINOv2$_f$ |
  | --- | --- | ---- | --- | --- | --- | --- | --- |
   AR @ max10      | 26.24 | 26.33 | 23.55 | 25.82 | 29.84 | 31.25 | 40.02
   AR @ max100    | 39.24 | 49.38 | 44.72 | 46.28 | 54.22 | 63.05 | 63.06
   AR$_s$ @ max100  | 14.83 | 22.04 | 17.84 | 22.09 | 32.00 | 31.65 | 31.11
   AR$_m$ @ max100 | 44.87 | 56.76 | 52.03 | 52.85 | 59.43 | 70.01 | 70.40
   AR$_l$ @ max100   | 60.05 | 69.69 | 64.58 | 64.11 | 72.92 | 90.63 | 90.36

---

### Author Response · Authors · 2023-08-28
**final check if reviewers have additional concerns or comments**

We sincerely thank all the reviewers again for the helpful suggestions and insightful feedback. We hope our revision and responses properly address reviewers’ concerns. We would like to check if there are additional concerns or comments, and we are happy to provide further discussions, clarity or ablations.

---

### Author Response · Authors · 2023-08-30
**summary of paper, reviews, and rebuttal**

Dear Reviewers, and ACs,

We sincerely thank the reviewers again for their constructive suggestions and insightful feedback, and ACs' hard work of overseeing the reviewing process. Our submission now receives ***2x "7: Good paper, accept", and 1x "6: Marginally above acceptance threshold".*** Yet, we would like to summarize our submission to help you make the final decision, because the reviews, rebuttal, and active author-reviewer discussions make the openreview threads go too long for you to quickly grasp the state of our submission.

**Summary of our paper**. Our submission introduces a new dataset to foster the research of Instance Detection. As noted by all the reviewers, our dataset is novel, large-scale, clean, useful, and important. Reviewers appreciate our standardization effort in terms of dataset construction and evaluation protocol design, as well as the detailed documentation. We also introduce a non-learned method to approach Instance Detection, and reviewers find it to be novel and simple.

**Summary of reviews and rebuttal**. Reviewers give positive ratings, and provide constructive suggestions and insightful feedback. Major  suggestions/concerns are (1) adding an illustrative figure to help readers understand Instance Detection versus Object Detection, (2) adding recall metrics for in-depth analysis, (3) adding breakdown analysis w.r.t object size. In the rebuttal, we have effectively addressed these concerns / suggestions, and revised our paper. As a result, *all the reviewers have upgraded the ratings, which are 2x "7: Good paper, accept" and 1x "6: Marginally above acceptance threshold".*





Regards,

Authors of Paper-177

---

### Decision · Program_Chairs · 2023-09-22

**Decision:**

Accept (Poster)

**Comment:**

This paper receives universally positive reviews. The reviewers are excited about both the newly created dataset and the empirical studies from the simple baselines proposed by the paper. The ablations offer interesting perspectives to the problem. AC believes that the paper adds a significant contribution to the community to enable further research in the Instance Detection task.